# A Low-Profile Dielectric Resonator Antenna Array for OAM Waves Generation at 5G NR Bands

**DOI:** 10.3390/mi14040841

**Published:** 2023-04-13

**Authors:** Nur Akmal Abd Rahman, Shehab Khan Noor, Imran Mohd Ibrahim, Mohd Najib Mohd Yasin, Arif Mawardi Ismail, Mohamed Nasrun Osman, Shaiful Bakri Ismail

**Affiliations:** 1Advanced Communication Engineering (ACE) Centre of Excellence, Faculty Electronic Engineering Technology, Universiti Malaysia Perlis (UniMAP), Kangar 01000, Malaysia; 2Centre for Telecommunication Research and Innovation (CeTRI), Faculty of Electronic and Computer Engineering, Universiti Teknikal Malaysia Melaka (UTeM), Durian Tunggal 76100, Malaysia; 3Universiti Kuala Lumpur Malaysian Institute of Marine Engineering Technology (UniKL-MIMET), Dataran Industri Teknologi Kejuruteraan Marin, Bandar Teknologi Maritim, Jalan Pantai Remis, Lumut 32200, Malaysia

**Keywords:** dielectric resonator antenna, orbital angular momentum, uniform circular array, 5G communications, 5G New Radio (NR)

## Abstract

This paper presents the generation of orbital angular momentum (OAM) vortex waves with mode +1 using dielectric resonator antenna (DRA) array. The proposed antenna was designed and fabricated using FR-4 substrate to generate OAM mode +1 at 3.56 GHz (5G new radio band). The proposed antenna consists of 2 × 2 rectangular DRA array, a feeding network, and four cross slots etched on the ground plane. The proposed antenna succeeded in generating OAM waves; this was confirmed by the measured radiation pattern (2D polar form), simulated phase distribution, and intensity distribution. Moreover, mode purity analysis was carried out to verify the generation of OAM mode +1, and the purity obtained was 53.87%. The antenna operates from 3.2 to 3.66 GHz with a maximum gain of 7.3 dBi. Compared with previous designs, this proposed antenna is low-profile and easy to fabricate. In addition, the proposed antenna has a compact structure, wide bandwidth, high gain, and low losses, thus meeting the requirements of 5G NR applications.

## 1. Introduction

At present, wireless technologies with high-speed internet connectivity and higher data rates are in high demand. However, current wireless technologies belonging to the third generation (3G) and fourth generation (4G) cannot meet the demands of fifth-generation (5G) wireless requirements. Moreover, they cannot be used for long-distance communication and low-power, wide-area technology. Thus, 5G technology has become a most challenging and interesting topic in wireless research and will open up new opportunities to overcome existing development limitations. 5G New Radio (5G NR) is a new air interface defined by the 3rd Generation Partnership Project (3GPP) in Release 15 and consecutive releases [1]. It is specifically developed for future 5G operations. Recently, the 3GPP has released a new specification and has permitted the following new spectrums for future 5G NR working below the Sub-6 GHz frequency bands, namely Band n77, Band n78, and Band n79 [2]. Band n77 ranges from 3.3 to 4.2 GHz and is currently being used in the United States. Most Asian and European countries presently use Band n78, which ranges from 3.3 to 3.8 GHz. Meanwhile, Band n79 ranges from 4.4 to 5 GHz. The main advantages of the 5G are higher data rates, lower latency, massive capacity, and higher quality of service [3,4].

The electromagnetic (EM) momentum includes linear and angular momentum. Angular momentum consists of spin angular momentum (SAM) and orbital angular momentum (OAM), which define its polarization state and phase structure distribution, respectively [5]. SAM has been used for decades [6,7,8,9,10,11,12], while OAM was discovered by Allen et al. in 1992 [13]. The authors combined the concept of OAM with the idea of an optical vortex. In an optical vortex, the planes of the constant phase of the electric and magnetic vector fields form a corkscrew or helicoid running in the direction of propagation. Since then, OAM has been widely developed in the optical and quantum domains [10,11,12].

OAM is carried by EM waves, which have the helical phase factor of exp (*ilφ*), where *l* is the OAM mode number and *φ* is the transverse azimuthal angle [13]. For SAM, the rotation of the *E* and *H* fields is around the main beam axis. However, for OAM, the rotation of the *E* and *H* fields is around the propagation direction [14]. The existence of the phase distribution exp (*i**l**Φ*) indicates that on a plane perpendicular to the propagation axis, the phase of an OAM wave changes continuously by 2*π**l* upon one complete cycle around the beam axis. In addition, an OAM wave exhibits a helical wavefront, with *l* representing the number of wavefront twists within one wavelength.

OAM has several distinct advantages compared to SAM; for example, it can provide an infinite number of states or modes since the OAM mode number *l* is a limitless integer. Circularly polarized (CP) waves carry SAM with only two orthogonal states: either right-hand circular polarization (RHCP) or left-hand circular polarization (LHCP) [15]. In addition, the beams with different OAM modes are orthogonal to each other, and they can be multiplexed or demultiplexed together [16]. Due to the orthogonality, the spectrum efficiency and channel capacity can be increased. Hence, more users can use the same frequency channel without extending the bandwidth [17,18,19].

For OAM applications, various antennas have been designed and optimized. Various methods can be used to generate OAM modes, such as spiral phase plate (SPP) [20,21,22], parabolic reflector antenna [23,24], metasurface [25,26], and uniform circular array (UCA) antenna [27,28,29,30,31]. The fundamental principle of SPP is based on changing the output phase of the beams in proportion to the azimuthal angle around the center according to the OAM mode number. The SPP is widely used because of its simple structure. However, this method can only generate one OAM state once the structure is decided. Moreover, the dielectric plate with high weight generally brings additional losses and reduces the antenna efficiencies. Another popular method to generate OAM waves is through the concept of a parabolic reflector antenna. Parabolic reflector antenna offers high mode purity, gain, and directivity. Moreover, the design is simple. However, there are several disadvantages such as its limitations in generating multiple OAM modes and the feeder blocking the signal. Moreover, the parabolic antenna is difficult to fabricate and integrate with other elements, and its efficiency is low.

Metasurface is a promising candidate for generating OAM waves due to its unique advantages such as its artificial material, ease of fabrication, absence of complex feed networks, and capability of designing different scales according to the application. The metasurface controls the wavefront of EM waves by adjusting the phase of the incoming waves to generate OAM beams. However, most metasurface designs have large dimensions, which are not suitable since most current devices are compact. In addition, an additional resource such as a horn antenna is required to transmit the incident waves, and this antenna is placed in front of the metasurface, which can block the reflected OAM waves. The OAM waves can also be generated by using the concept of UCA antenna. UCA is widely used due to its advantages such as great adaptability, capacity to generate multiple OAM modes, compact size, and low cost. A UCA antenna consists of equal antenna elements which are organized uniformly in a circle. The radiating elements are excited with the power of the same amplitude but in different phases. This method can generate multiple OAM modes, and the purity of the OAM mode can be easily increased. However, the spacing among the adjacent radiating elements must be at least 0.7λ to prevent mutual coupling; thus, the antenna tends to be bigger at lower frequencies. Moreover, the misalignment of radiating elements will distort the spiral phase distribution.

Recently, numerous studies have been carried out to generate OAM waves using a dielectric resonator antenna (DRA) [32,33]. DRAs have several advantages such as compact size, low cost, high radiation efficiency, and flexibility toward various feed lines. Moreover, DRAs have low loss compared to microstrip patch antenna since they do not have any metallic surfaces such as radiating elements. OAM waves can be generated by using different types of DRAs, such as array, cylindrical, and hemispherical. However, not many works present the generation of OAM vortex waves with mode +1. In [32], the authors designed a water spiral DRA to generate multimode OAM waves. The shell of the spiral DRA filled with distilled water is designed as a cylinder with a spiral paraboloid on the top side and fed with two orthogonal signals at the bottom side to excite higher-order modes. By changing the radius, height, and notch height of the spiral DRA, *l* = ±1 OAM modes can be generated at 4.8 and 5.6 GHz, and *l* = ±3 OAM modes can be generated at 5.11, 5.66, and 6.36 GHz, respectively. Although the measurement results proved that OAM waves were generated based on the spiral phase distribution, the purity of the generated OAM mode was not discussed. In [33], an antenna array with 8 slots and coupled cylindrical DRAs fed by microstrip lines was proposed to generate mode +1 and mode +2 at 3.5 GHz. However, the obtained gain value and mode purity were not disclosed in the paper.

To the authors’ best knowledge, this paper is unprecedented in proposing a rectangular DRA with UCA-based OAM characteristics which is specifically designed for 5G NR applications. Section 2 elucidates the antenna design and its operation mechanism. The effect of array arrangement on compactness is also discussed in this section. The experimental analysis of the proposed antenna in terms of reflection coefficient, radiation pattern, gain, phase distribution, 2D polar form, intensity distribution, and mode purity are discussed in Section 3. Finally, Section 4 presents our conclusions.

## 2. Design Methodology

The geometry of the proposed DRA is illustrated in Figure 1. Four rectangular DRs are placed on top of a feeding structure. The DR is made of microwave dielectric ceramics ($Al_{2}O_{3}$) with a dielectric constant $\varepsilon_{r}\;$ of 10. The size of the DR is $W_{dr}\;$×$\;L_{dr}\;$×$\;h_{dr}$ = 22 mm × 22 mm × 32 mm. The proposed DRA is designed and fabricated on an FR-4 substrate with a dielectric constant of 4.3, loss tangent tan *δ* of 0.025, and thickness *h* of 1.6 mm. The overall dimensions of the proposed antenna operating at 3.5 GHz are 185 mm × 160 mm (2.16λ × 1.87λ). The proposed DRA consists of a copper ground layer with a thickness of 0.035 mm. Four crossed-shaped slots with equal dimensions are inserted in the ground plane for the single-band operation.

For this work, a UCA with four elements was designed. For an ***N***-element UCA, an OAM wave of topological charge *l* can be generated by exciting the elements with the power of equal amplitude, with a phase difference of ΔØ=2πl/N between adjacent elements. Hence, to generate OAM mode ±1 with four dielectric resonators, the phase difference must be 90° between adjacent antenna elements. The dielectric resonators are placed in a circular arrangement with a radius (*R*) of 77 mm. The array radius must be at least 0.7λ to avoid mutual coupling and shift in the resonant frequency. The proposed antenna was designed with a feeding network, as shown in Figure 1b. The feeding network was used to excite the dielectric resonator by supplying similar output energy at output ports with the required phase shift value. Once the antenna was designed and fabricated, parameters such as reflection coefficient, radiation pattern, phase distribution, and intensity distribution were measured with the help of a vector network analyzer and a horn antenna inside an anechoic chamber at Advanced Communication Engineering (ACE) Universiti Malaysia Perlis. After the measured data were collected, Python programming was used for data analysis.

The proposed antenna is excited by a single source. Equal power distribution to the elements is realized using a feeding network with T-junction dividers. The desired phase difference between adjacent elements is achieved by optimizing the lengths of the feed lines and the geometry of the feeding network. The feeding network of UCA is shown in Figure 2. Port 1 is the input port, whereas port 2, port 3, port 4, and port 5 are the output ports used to excite DR 1, DR 2, DR 3, and DR 4, respectively. A 90° phase difference between ports 2 and 3 was obtained by altering the length of the feedline, as illustrated in Figure 2b. Ports 2 and 3 are the mirror image of ports 4 and 5. The mirror image arrangement of the feeding network was adopted for the simplicity of the design. The overall phase difference for port 2, port 3, port 4, and port 5 were −97.85°, −0.58°, −93.36°, and −0.22°, respectively, as shown in Figure 3a. The additional 180° phase shift due to the mirror image of the feeding network gives port 4 a phase shift of 86.64° and port 5 a phase shift of 179.78°. Thus, the incremental phase shifts for the proposed antenna were −97.85°, −0.58°, 86.64°, and 179.78°. The phase differences were as follows: 97.27° between ports 2 and 3; 87.22° between ports 3 and 4; 93.14° between ports 4 and 5; and 98.07° between ports 5 and 1. It has been observed from the simulated result that the obtained values are close to 90°. Moreover, the amplitudes for all four ports were very similar, as shown in Figure 3b. The amplitudes for port 2, port 3, port 4, and port 5 were −9.41 dB, −10.84 dB, −9.23 dB, and −10.46 dB, respectively. The dimensions of all parameters are well optimized to resonate at 3.2 to 3.66 GHz, and the details of the parameters are shown in Table 1.

The fabrication and measurement processes took place once all the simulation results displayed good agreement with the design goal and the antenna’s requirements. The proposed DRA was fed with a 50 Ω SMA connector. An SMA connector was used because it offers the best impedance matching, given that it can be adjusted to any position. The fabricated prototype of the proposed DRA is shown in Figure 4.

## 3. Results and Discussion

The successful generation of OAM vortex waves was verified by analyzing its spiral phase distribution [27,28,29], a donut-shaped intensity distribution [32], a null in the center of the radiation pattern [31], and the purity percentage of the chosen mode should be higher than the other modes carried by the OAM waves.

### 3.1. Reflection Coefficient ($S_{11}$)

The measurement setup to measure the reflection coefficient ($S_{11}$) and bandwidth of the proposed antenna is shown in Figure 5a. A Keysight Technologies E5071C E-Series vector network analyzer was employed to carry out the measurement. This type of vector network analyzer is an integrated network analyzer with two or four ports and an S-parameter test set and covers a wide frequency range, from 300 KHz to 20 GHz. The simulated and measured reflection coefficient for the proposed antenna is shown in Figure 5b. It was observed that the measured reflection coefficient values range from 3.2 to 3.66 GHz, which are slightly greater than the simulated reflection coefficient values. Nevertheless, the proposed design has an acceptable reflection coefficient value and at least 80 to 100 MHz bandwidth as these are the conditions of 5G NR band deployment.

### 3.2. Phase and Intensity Distributions

For experimental validation, the phase and intensity distributions of the proposed antenna were performed in an anechoic chamber. The measurement setup is shown in Figure 6. As shown in Figure 6, a 2-axis linear guide rail and a receiving antenna were used to scan the phase over a sampling plane that is perpendicular to the propagation of the radiation produced by the transmitting antenna. To ensure consistency of results, the transmitting antenna was placed at a constant distance, *D* of 1.2 m from the receiving antenna. The positioning of the receiving antenna on the linear guide rail was controlled by a computer with a suitable step size. The signal measured by the receiving antenna is sent to a vector network analyzer to determine the phase, and the data are then stored in a file on a computer. The data obtained were used to plot the phase distribution and were analyzed with Python to determine the purity of the OAM waves.

The simulated phase distribution of the proposed antenna from 0° to 360° is shown in Figure 7a. It was proven that the proposed antenna exhibited a clockwise spiral phase distribution with a continuous phase change of 2π around the center. This is a characteristic feature of the OAM wave. The multicolor in Figure 7a indicates that the phase change has taken place by 2π around the propagation axis. In addition, the measured phase distribution also exhibited a spiral phase distribution, as shown in Figure 7b. As shown in Figure 7c, a hollow structure that resembles a donut shape which is called a “vortex core” can be observed at the center of the beam. This black dot inside the blue circle shape of the beam determines the intensity distribution of the produced OAM mode. The measured intensity distribution is shown in Figure 7d. As stated in [32], the OAM vortex beams with mode ≠ 0 have a donut-shaped structure, and this donut-shaped structure expands in dimensions as the value of OAM modes increases. Therefore, both spiral phase distribution and donut-shaped intensity were evident from the simulated results. These confirm the generation of OAM mode +1 using the proposed design.

### 3.3. Radiation Pattern

The experimental setup to measure the 2D radiation patterns in the *E*-plane of the DRA is shown in Figure 8a. The proposed antenna acted as the transmitting antenna or antenna under test (AUT), while the horn antenna acted as the receiving antenna. This measurement was conducted inside a fully equipped anechoic chamber with the assistance of a vector network analyzer and Diamond Engineering Automated Measurement Systems (DAMS). The radiation pattern setup for the proposed DRA is shown in Figure 8b.

The simulated and measured 2D polar radiation pattern at phi = 0 (*E*-plane or *xz* plane) is shown in Figure 9a. Both the simulated and measured radiation patterns show very similar performance for the proposed DRA. A null at the center of the radiation pattern indicates a characteristic of OAM waves, which is apparent in the plot. However, there are some differences between the simulation and measurement results due to discrepancies in the fabrication process. Moreover, it can be observed that the measured beamwidth is narrower than the simulated beamwidth. The simulated 3D radiation pattern is shown in Figure 9b, where a null at the center of the beam can be observed clearly. Gains of between 3.6 dBi and 7.3 dBi are observed across these frequencies. The slight differences between the simulated and measured results could have occurred due to inaccuracies during fabrication and experimental tolerances.

### 3.4. Mode Purity

Generating OAM waves with mode purity is crucial to make sure the modes are orthogonal to one another. Mode purity is the main parameter to measure the consistency of information transmission using OAM waves. Several factors can be considered to generate high purity of OAM modes, such as the number of elements, transmitting antenna size, receiver size, and observation plane size [34,35].

In [36], the author verified the importance of a higher number of array elements in mode purity. Ripples will occur in the radiation pattern if the number of array elements is not enough to generate the desired OAM mode. The minimum number of elements required to generate a specific OAM mode can be found using the following equation:(1)$$N\geq 2\;\left|l\right|+1$$
where *N* is the number of array elements and *l* is the mode number. As the number of elements in the array increases, the phase difference between the adjacent elements will be smaller, thus reducing the phase ambiguity effect. However, increasing the number of elements in the array will increase the size of the antenna, and the design will become more complex. Hence, special care during the simulation process needs to be taken to determine how many elements are required for a certain OAM mode to confirm that the mode has high purity. Another factor that can influence mode purity is the transmitting array antenna size and observation plane [36]. Decreasing the radius of the array will increase the mode purity. However, there is still the problem of mode divergence.

In this paper, the phase distribution obtained from the CST simulation is used to analyze the mode purity by using Python programming. The mode purity of the OAM beams is calculated by using Fourier transform, as given by the following equations:(2)$$\Psi \left(Ø\right)=\frac{1}{\sqrt{2\pi }}\overset{\infty }{\underset{l=-\infty }{\sum }}A_{l^{e^{jl\emptyset }}}$$
(3)$$A_{l}=\frac{1}{2\pi }\int_{-\pi }^{\pi }\Psi \left(Ø\right)e^{-jl\emptyset }d\emptyset$$
where Ψ(Ø) is the angular distribution of the EM wave and $A_{l}$ is the distribution of angular momentum of different OAM modes. Generally, generated OAM waves consist of multiple OAM modes. The proposed DRA generated OAM mode +1 with a purity of 53.87%, as shown in Figure 10. The remaining 46.13% is distributed over other adjacent modes in the spectrum. The antenna was designed to produce OAM mode +1; hence, the ratio of purity is more for mode +1 when compared to the OAM wave as a whole.

Table 2 compares the performance of the proposed design with some previous works on OAM antennas in terms of frequency, antenna dimension, gain, mode number, mode purity, and measurement of phase and intensity distributions. It is evident that the design of this paper has a small size, wide bandwidth, and high gain. Moreover, the proposed antenna manages to generate OAM at mode +1 with high mode purity. From the literature, it can be seen that all of the previous works disregarded the OAM mode purity information.

## 4. Conclusions

A rectangular DRA with an inset-fed technique and uniform circular array configuration was designed to generate OAM mode +1 at 3.5 GHz for the 5G NR band. Moreover, a mirror image arrangement of the feeding network was adopted to make the structure simple and less complex. The proposed antenna indicates that the dielectric resonator element has low metallization and can increase the gain and radiation efficiency of the antenna. In addition, the array DRA with a cross slot at the ground plane manages to enhance the bandwidth of the antenna. Moreover, the implementation of the dielectric resonator successfully contributes to antenna miniaturization, yielding its compact size of 2.16λ × 1.87λ. To further validate the performance of the proposed design, the 2D far-field radiation patterns are measured using a standard horn antenna, and the measured results coincide well with the simulated ones. The measured results show that the antenna arrays can cover the 3.5 GHz band of the 5G NR spectrum. The electric field phase patterns confirm the generation of OAM beams. The proposed antenna generated an OAM mode of high purity and achieved a wide bandwidth of 460 MHz with a purity of 53.87%. Future work could include different shapes of the dielectric resonator to increase the bandwidth of the antenna. In addition, a modified slot in the ground plane could also be used to achieve wide bandwidth performance. Using the different shapes of the slot is advantageous as it simplifies the process. In comparison with previous studies, this method is believed to be a significant development in producing a compact size, simple structure, and wide bandwidth to generate OAM waves at the 5G NR band. Lastly, the mode purity analysis of this work shall guide future researchers and engineers in selecting the receiver size of the OAM wave-based wireless communication system.

## Figures and Tables

**Figure 1 micromachines-14-00841-f001:**
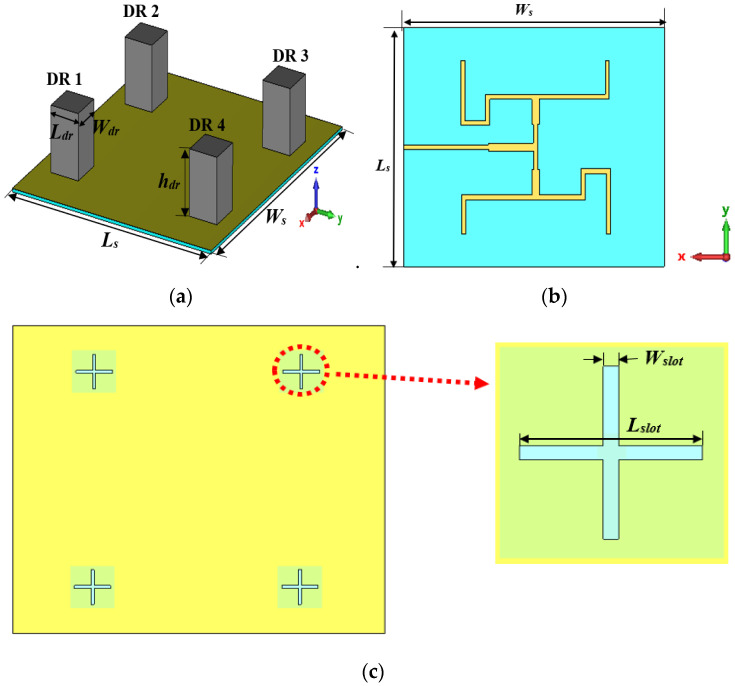
Proposed DRA. (**a**) Perspective view; (**b**) back view; (**c**) ground plane.

**Figure 2 micromachines-14-00841-f002:**
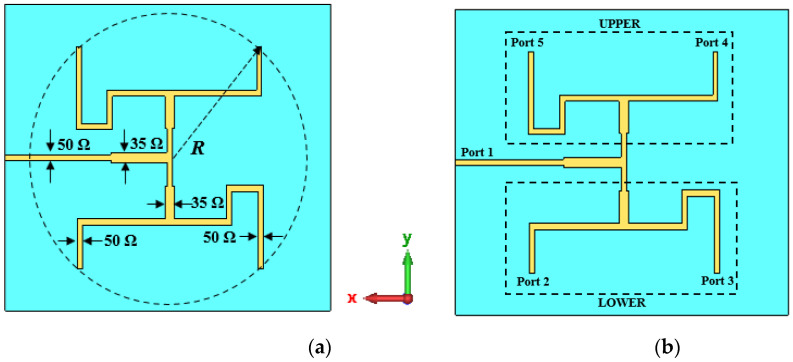
Feeding network. (**a**) Structure of the feeding network; (**b**) location of the ports.

**Figure 3 micromachines-14-00841-f003:**
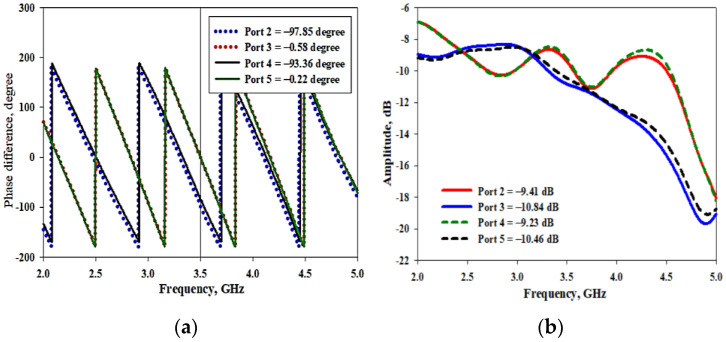
Simulated results of the feeding network. (**a**) Phase shifts between the elements; (**b**) amplitude.

**Figure 4 micromachines-14-00841-f004:**
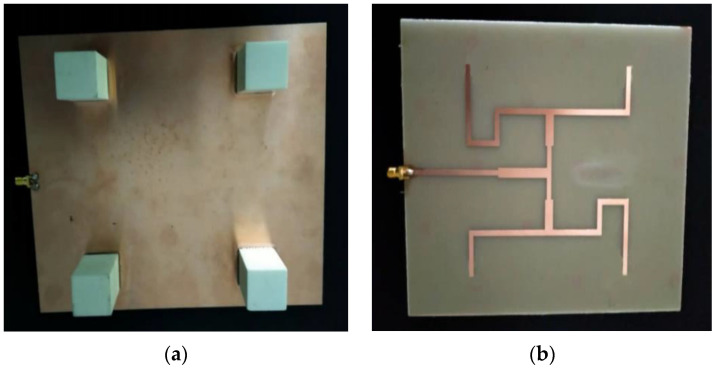
Fabricated prototype. (**a**) Front view; (**b**) back view.

**Figure 5 micromachines-14-00841-f005:**
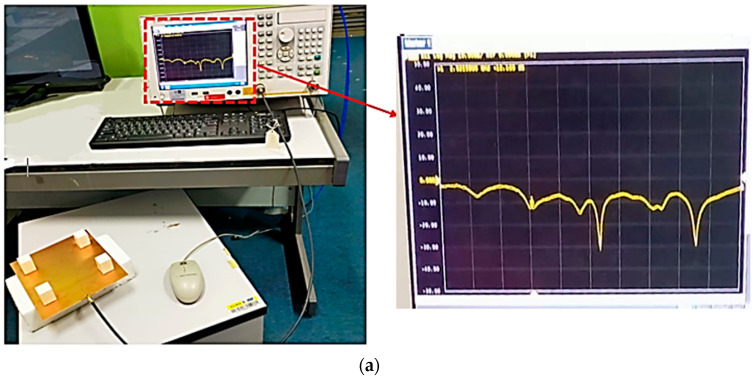
Reflection coefficient of the proposed antenna. (**a**) Measurement setup; (**b**) simulated and measured results of the reflection coefficient.

**Figure 6 micromachines-14-00841-f006:**
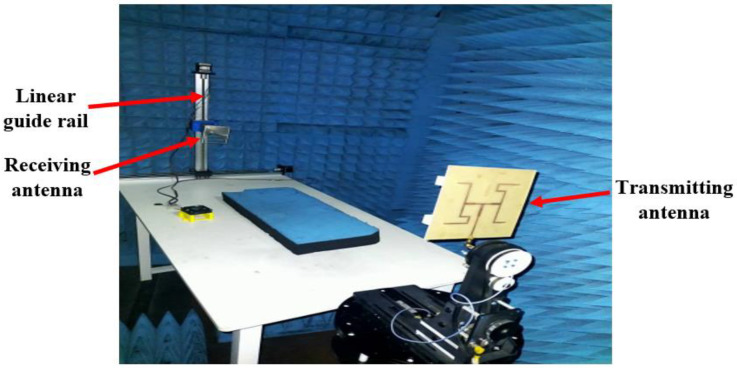
Measurement setup of phase and intensity distributions.

**Figure 7 micromachines-14-00841-f007:**
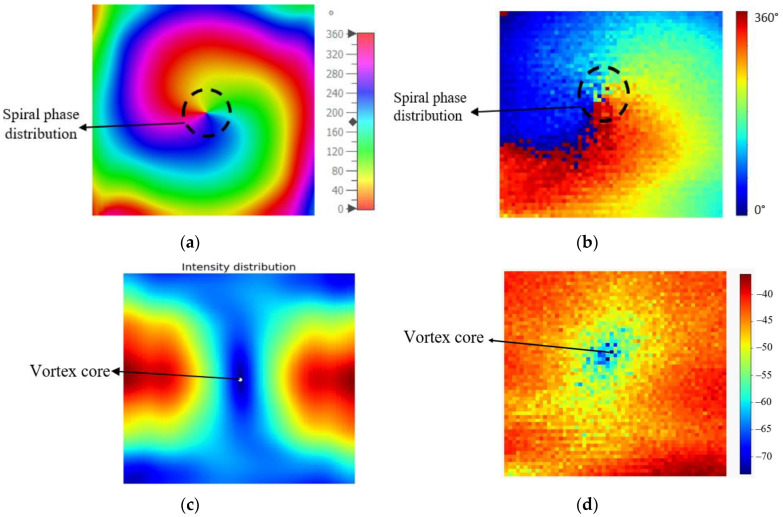
Radiation and intensity distributions at 3.56 GHz. (**a**) Simulated phase distribution; (**b**) measured phase distribution; (**c**) simulated intensity distribution; (**d**) measured intensity distribution.

**Figure 8 micromachines-14-00841-f008:**
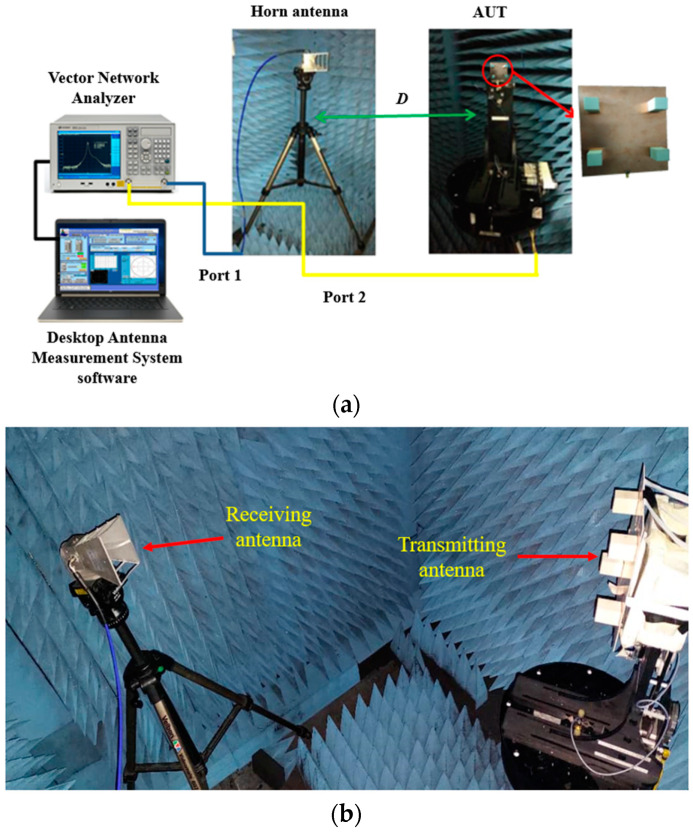
2D radiation pattern measurement setup. (**a**) Overall graphical presentation of the measurement setup; (**b**) DRA setup inside the anechoic chamber.

**Figure 9 micromachines-14-00841-f009:**
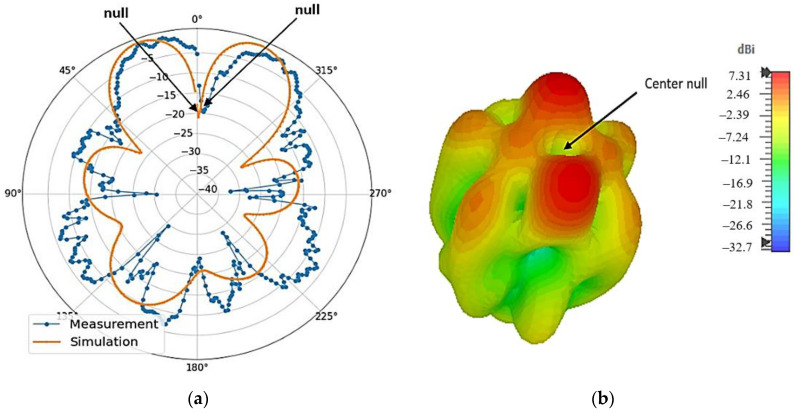
Far-field results at 3.56 GHz. (**a**) Simulated and measured 2D radiation patterns in *E* plane; (**b**) simulated 3D radiation pattern.

**Figure 10 micromachines-14-00841-f010:**
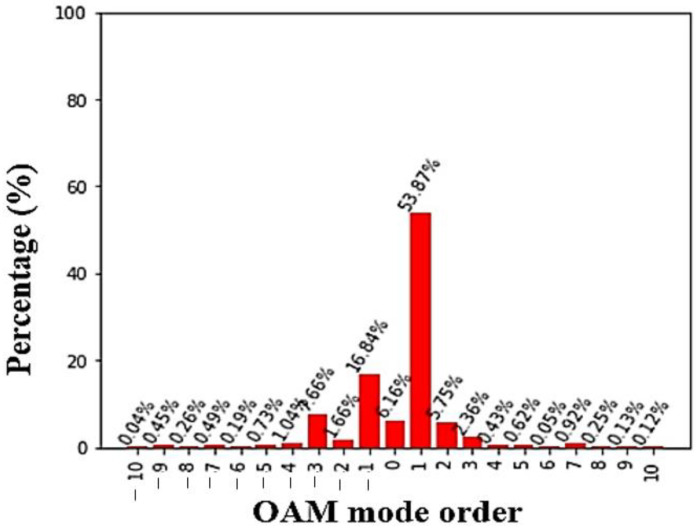
Mode purity of OAM mode +1 at 3.56 GHz.

**Table 1 micromachines-14-00841-t001:** The dimensions of the proposed DRA.

Parameters	Values (mm)	Parameters	Values (mm)	Parameters	Values (mm)
$L_{s}$	160	$W_{dr}$	22	$L_{slot}$	36
$W_{s}$	185	$h_{dr}$	32	$W_{slot}$	1.5
$L_{dr}$	22	*h*	1.6	*R*	77

**Table 2 micromachines-14-00841-t002:** Comparison between various OAM antennas.

Ref.	*f_r_* (GHz)	Imp. BW (%)	Antenna Dimension	Gain (dBi)	Mode Number	Mode Purity (%)	Measurement of Phase and Intensity Distributions
[25]	4.5–7	43.5	5.83λ × 5.83λ	NA	+1 and +2	NA	Yes
[26]	4.9–6.5	28.1	1.4λ × 1.4λ	15.19	0 and ±1	NA	Yes (Phase only)
[27]	5.72–5.95	3.94	*R* = 1.46λ	NA	±1	NA	Yes (Phase only)
[28]	1.54–1.56	1.29	NA	NA	±1	NA	No
[29]	2.25–2.4	6.45	3.5λ × 3.5λ	NA	+1 and +2	NA	Yes (Phase only)
[30]	10.2–10.7	4.78	NA	4.6	−1	NA	No
[31]	9.25–10.5	12.66	*R* = 0.19λ	NA	+1	NA	Yes (Phase only)
[32]	5.07–5.14	1.37	*R* = 0.37λ	1.08	±1 and ±3	NA	Yes (Phase only)
[33]	2.8–4.55	47.6	NA	NA	+1 and +2	NA	No
This work	3.2–3.66	13.4	2.16λ × 1.87λ	7.3	+1	53.87	Yes

## Data Availability

Not applicable.

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
