# Peer review of "A Low-Profile Dielectric Resonator Antenna Array for OAM Waves Generation at 5G NR Bands"

_micromachines, 2023, doi:10.3390/mi14040841_

Round 1

Reviewer 1 Report

The paper needs extensive grammatical revision before it can be accepted. Otherwise, the antenna design, application, and results are interesting and well presented. I also encourage the authors to improve figure resolution before resubmitting.

Reviewer 2 Report

A DRA array was designed and fabricated to generate OAM +1 mode waves. Furthermore mode purity analysis was carried out to verify the generation of OAM mode +1. However, some questions and comments are as follows:

1- In Lines 54, 111, 112, and 136, the " l " should be in italic, to avoid confusion with 1(one).

2- Ports 2 and 3 are the mirror image of ports 4 and 5. Therefore, it is expected that the simulated phase difference for port 4 and port 5 are the same as port 2 and port 3, respectively.  Your simulation doesn't show it (the phase differences are -97.85°, -0.58°, -93.36°, and -0.22°). What's the reason? This problem also exists for the amplitude simulation. (The amplitude for port 2, port 3, port 4, and port 5 are -9.41 dB, -10.84 dB, -9.23 dB, and -10.46 dB, respectively.)

3- I think simulation results in figure 3 are the feeding simulation results without the cross shaped slots and DRs. Please mention this issue in the article. Furthermore, add simulated S11 results of the feeding network to figure3.

4- In the frequency range of 3.7 to 5 GHz, the amplitude curves in Figure 3 decrease rapidly. What is your explanation about the rapid reduction in output powers?

5- The vertical color axis in figure 7b should be in the rage of 0 to 360 degrees, as in figure 7a.

6- As can be seen in figure 3b, the feeding network has significant loss, which reduces the antenna efficiency. Therefore, contrary to what you claimed in the Abstract, the designed antenna is not a high efficiency antenna. Pleas report the efficiency of the proposed antenna.

7- For better comparison in Table 2, its better that the antenna dimensions are report in terms of wavelength.

8- It is not usual to refer to articles in the Conclusion.

Reviewer 3 Report

This paper describes an experimental demonstration of an OAM antenna with a high mode purity. A very high purity has been experimentally achieved over existing references. However, scientific discussion about the performance is poor. So, I recommend adding a sufficient discussion as a minor revision.

1. Although the achieved purity value of 54% is high, its discussion is poor. Please describe how high purity is expected through the simulation, and compare it with the experimental result. In addition, a description of how to improve the purity (i.e. increasing N or something like that) is required.

2. In figs 6 and 7, please describe the value and effect of the distance between the antenna and the detector.

3. In fig. 7c and d, the intensity distribution is not a donut shape. Why?

4. In line 214, port four and five should be port 4 and 5.

5. Please rotate fig. 4b to meet to other figures.
